# UniFork: Exploring Modality Alignment for Unified Multimodal Understanding and Generation

## Abstract

Unified image understanding and generation has emerged as a promising paradigm in multimodal artificial intelligence. Despite recent progress, the optimal architectural design for such unified models remains an open challenge. In this work, we start by analyzing the modality alignment behaviors of task-specific expert models for understanding and generation, as well as current unified models. Our analysis reveals a crucial observation: understanding tasks benefit from a progressively increasing modality alignment across network depth, which helps build up semantic information for better comprehension; In contrast, generation tasks follow a different trend—modality alignment increases in the early layers but decreases in the deep layers to recover spatial details. These divergent alignment patterns create a fundamental conflict in fully shared Transformer backbones, where a uniform representational flow often leads to performance compromises across two tasks. Motivated by this finding, we introduce **UniFork**, a novel Y-shaped architecture that shares the shallow layers for cross-task representation learning, while employing task-specific branches in deeper layers to avoid task interference. This design effectively balances shared learning and task specialization. Through extensive ablation experiments, we demonstrate that Unifork consistently outperforms conventional fully shared Transformer architectures, and achieves performance on par with or better than task-specific models.

## 1 Introduction

Recent works (Xie et al., 2024; Li et al., 2025a; Deng et al., 2025; Zhang et al., 2025) have demonstrated significant progress in unified multimodal generation and understanding. By projecting both language and vision signals into a shared embedding space and arranging them in various ways, it becomes feasible to perform both image understanding and generation within a single Transformer architecture. However, despite sharing such a common paradigm, the objectives of generation and understanding tasks are inherently different (Wu et al., 2025; Chen et al., 2025b). Image generation emphasizes the fidelity and aesthetic quality of visual outputs, focusing on pixel-level details such as texture and color. In contrast, image understanding centers on high-level semantic comprehension, such as identifying objects, interpreting spatial relationships, and reasoning about scene content. This fundamental divergence makes it notoriously challenging to unify the two tasks .

To address the task discrepancy issue, some recent approaches (Wu et al., 2025; Chen et al., 2025b) adopt distinct semantic and spatial image representations tailored to understanding and generation respectively. Other methods introduce diffusion optimization objectives (Xie et al., 2024; Zhou et al., 2024) or external models (Ge et al., 2024a; AI et al., 2025) to decode spatial features for image generation. Although these designs can enhance task-specific performance, they often undermine the simplicity and elegance of the original next-token prediction (NTP) paradigm in large language models (LLM). In addition, during supervised fine-tuning (SFT), meticulous data balancing is typically required to maintain performance across tasks. Furthermore, the intrinsic relationship between generation and understanding remains largely unexplored, raising important questions about how these tasks might complement each other within a unified framework.

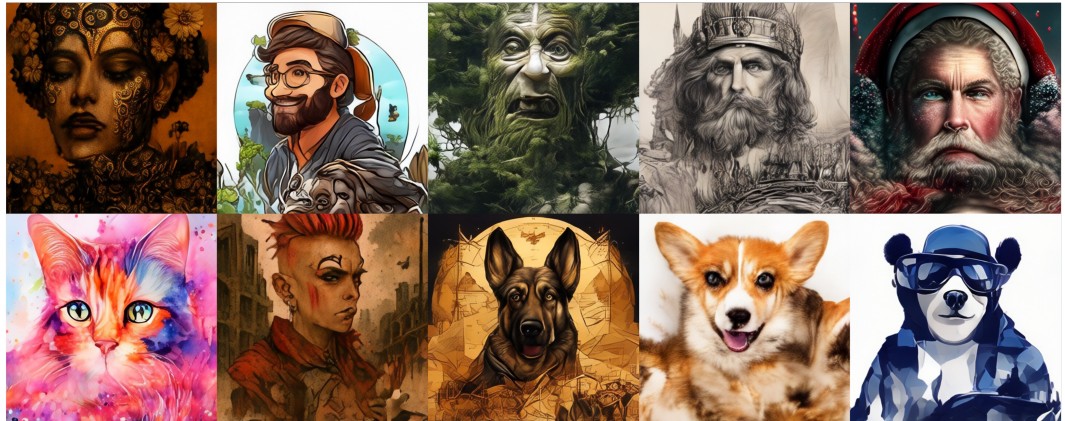

Figure 1: Text-to-image generation results by UniFork in 384×384 resolution.

In this work, we investigate the relationship between image understanding and generation through the lens of feature alignment between image and language tokens. We find that these two tasks exhibit distinct alignment patterns: image understanding benefits from progressively increasing alignment across network depth to build semantic representations, whereas image generation relies on strong early-layer alignment followed by weakened coupling in later layers to enable fine-grained visual synthesis. Moreover, employing a fully shared Transformer backbone under the NTP modeling paradigm enforces a representational compromise between the two tasks. These findings underscore the importance of accounting for the divergent alignment patterns of understanding and generation when designing unified models, in order to achieve optimal performance across both tasks.

Building upon the observation, we propose **UniFork**, a Y-shaped architecture for unified image understanding and generation. Specifically, the early layers of the Transformer backbone are shared across both tasks to enable cross-task semantic learning. In the latter layers, we introduce task-specific branches—two structurally identical yet independently parameterized modules. The understanding branch refines semantic representations, whereas the generation branch reconstructs spatial details. By decoupling the task-specific representation learning in the later layers, UniFork effectively alleviates the representational conflict arising from divergent alignment patterns. An additional advantage of UniFork lies in its training flexibility. During the final SFT stage, task-specific parameters can be independently optimized using their respective datasets, eliminating the need for delicate data ratio adjust. To validate the effectiveness of our design, we conduct extensive ablation studies showing that UniFork outperforms fully shared architectures and achieves performance comparable to task-specific expert models. Furthermore, with moderate scaling based on Qwen2.5-0.5B LLM (Yang et al., 2025), UniFork outperforms the state-of-the-art unified models trained at similar scale. Our main contributions are summarized as follows:

- We analyze task-specific modality alignment patterns in expert models, highlighting the differing needs of image understanding and generation, and providing insights for unified model design.

- We propose UniFork, a Y-shaped architecture that decouples task-specific learning in the later layers while retaining shared semantic representation learning in the early layers. This design enables effective cross-task learning and alleviates performance conflicts between tasks.

- Comprehensive ablation studies demonstrate that UniFork outperforms fully shared Transformer architectures. With moderate scaling, our approach achieves significantly improved performance on both understanding and generation tasks.

## 2 RELATED WORK

**Visual Generation.** Mainstream visual generative models can be categorized into diffusion-based methods and autoregressive (AR) approaches. Diffusion models (Rombach et al., 2022b; Esser et al., 2024; Labs, 2024) typically encode images into a continuous latent and generate it by progressively denoising a sampled Gaussian noise. While these models excel at producing photorealistic images,

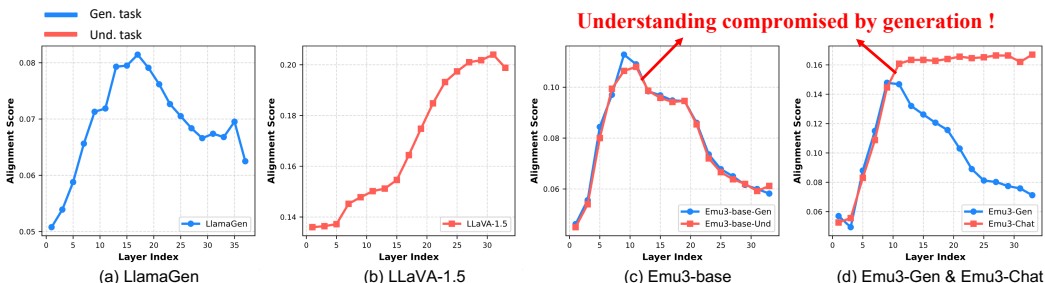

Figure 2: Modality alignment analysis. We visualize how text-image feature alignment evolves across Transformer layers for both image understanding and generation tasks: (a) Image generation exhibits a rise-then-fall alignment trend across layers. (b) Image understanding shows an increasing alignment pattern. (c) When using a fully shared Transformer for both tasks under the next-token prediction objective, the alignment curves converge, reflecting representational compromise between generation and understanding. (d) Models fine-tuned on Emu3-base (Wang et al., 2024) for each individual task recover their distinct trends, consistent with those observed in expert models.

the discrepancy between their continuous modeling of visual signals and the discrete token-based nature of language generation introduces significant architectural complexity when applied to unified multimodal frameworks. In contrast, AR models (Sun et al., 2024; Ramesh et al., 2021; Wang et al., 2025a) adopt a GPT-style generation paradigm by discretizing images into token sequences and generating them sequentially. This formulation naturally aligns with the LLMs, making these approachs more suitable for unified multimodal modeling. Representative models such as DALL·E (Ramesh et al., 2021) and LlamaGen (Sun et al., 2024) exhibit strong instruction-following capabilities and produce high-fidelity images. To further improve generation efficiency, recent works have introduced alternative generation paradigms, including next-scale generation (Tian et al., 2024), next-neighbor generation (He et al., 2025), and parallelized generation (Wang et al., 2025c).

**Unified Image Understanding and Generation.** Unified multimodal models (Wang et al., 2024; Team, 2024) aim to perform both visual understanding and generation within a single architecture, enabling the emergence of more advanced capabilities (Liao et al., 2025; Deng et al., 2025). However, previous image generation methods (Esser et al., 2024; Labs, 2024) mainly use diffusion-based frameworks with spatial autoencoders, while image understanding methods (Bai et al., 2025; Zhu et al., 2025) typically adopts an AR formulation with semantic encoding. This paradigm gap introduces practical challenges for unifying the two tasks within a single model. To bridge this gap, early approaches (Sun et al., 2023b; Wu et al., 2024a; Ge et al., 2024b) introduced external diffusion models for image generation. Other methods directly integrate diffusion objectives into the training of a shared Transformer backbone (Xie et al., 2024; Zhou et al., 2024), removing the need for separate diffusion heads. Representative frameworks such as the Janus series (Wu et al., 2025; Ma et al., 2025; Chen et al., 2025b) explicitly decouple visual encoding into dual pathways: a semantic encoder for understanding and a spatial encoder for generation. More recent works introduce task-specific (Wang et al., 2025b; Deng et al., 2025) or modality-specific (Li et al., 2025a) parameters within the LLM-based Transformer itself, distributing parts of the network to different tasks while maintaining a unified input-output interface.

While these designs improve performance on individual tasks, they often increase architectural and training complexity of the whole framework. Moreover, the relationship between visual understanding and generation remains underexplored, leaving open questions regarding the optimal structure.

## 3 METHOD

### 3.1 OBSERVATION AND ANALYSIS

Recent efforts have aimed to unify image understanding and generation within a single framework with various architectures. However, few works have examined the intrinsic relationship between the two tasks. In this study, we investigate their differences through the lens of modality alignment.

Given an image $\mathcal{X}$ and its corresponding textual prompt $\mathcal{T}$, we denote vision features extracted at the $l$-th Transformer layer as $V_l^{\text{gen}} \in \mathbb{R}^{n_v \times c}$ and the textual prompt feature from the final Transformer layer as $T \in \mathbb{R}^{n_t \times c}$. $n_v$ and $n_t$ represent the number of visual and textual tokens respectively, and $c$ is the channel size of the features. For the generation task, we sample 500 prompts from the GenEval (Ghosh et al., 2023) dataset. At each layer $l$, we compute the modality alignment score $A_l^{\text{gen}}$ using mutual k-nearest neighbors (mutual-kNN), a commonly used metric for evaluating representation alignment (Huh et al., 2024):

$$A_l^{\text{gen}} = \text{mutual-kNN}\left( \left\{ \frac{1}{n_v} \sum_{i=1}^{n_v} V_{l,i}^{\text{gen}}[b] \right\}_{b=1}^{500}, \left\{ \frac{1}{n_t} \sum_{j=1}^{n_t} T_j[b] \right\}_{b=1}^{500} \right).$$

For the understanding task, we feed the generated images into the model with the query "Provide a one-sentence caption for the image:". We extract the vision feature from each layer $V_l^{\text{und}}$ and compute the alignment score between $V_l^{\text{und}}$ and its corresponding prompt feature in each layer:

$$A_l^{\text{und}} = \text{mutual-kNN}\left( \left\{ \frac{1}{n_v} \sum_{i=1}^{n_v} V_{l,i}^{\text{und}}[b] \right\}_{b=1}^{500}, \left\{ \frac{1}{n_t} \sum_{j=1}^{n_t} T_j[b] \right\}_{b=1}^{500} \right).$$

**Divergent Alignment Patterns in Generation and Understanding.** Using this analytical tool, we begin by obtaining the alignment patterns in expert models (Sun et al., 2024; Liu et al., 2024b) trained separately for generation and understanding. As shown in Figure 2(a), we observe that in the generation task, the alignment score increases in early layers but decreases in later layers. This trend is consistent with observations from the REPA (Yu et al., 2024) study on diffusion models, suggesting that early layers focus on cross-modal alignment and semantic grounding, while later layers are responsible for synthesizing high-frequency visual details. In contrast, the understanding task exhibits an increasing alignment score across layers in Figure 2(b), indicating the importance of strong cross-modal alignment in deeper layers for accurate comprehension. These findings reveal that the two tasks require fundamentally different alignment behaviors.

**Representation Compromise of Fully Shared Backbones under NTP.** We then examine Emu3-base (Sun et al., 2023b), a native multimodal model pretrained jointly on both tasks. As shown in Figure 2(c), the alignment curves for generation and understanding nearly overlap, both following an increase-then-decrease pattern. This suggests that the understanding task may have compromised the generation objective during training. To validate this, we analyze two task-specific variants fine-tuned from Emu3-base: Emu3-Gen (Sun et al., 2023b) and Emu3-Chat (Sun et al., 2023b). Interestingly, as shown in Figure 2(d), Emu3-Chat recovers the monotonically increasing alignment trend characteristic in understanding tasks, while Emu3-Gen retains the rise-then-fall pattern typical of generation. This further supports our hypothesis that the two tasks prefer different alignment dynamics, and simply sharing a backbone under NTP paradigm may lead to representational conflict.

Motivated by these observations, we propose a Y-shaped architecture that shares early layers for joint semantic learning and decouples later layers to accommodate task-specific alignment needs.

## 3.2 ARCHITECTURE

The overall architecture of UniFork is illustrated in Figure 3, which enables both learning across tasks and task-specific specialization within a unified framework.

**Visual Tokenizer.** We adopt a single image tokenizer for both understanding and generation to maintain architectural simplicity. Our early exploratory experiments revealed that VAE-based tokenizers perform poorly under limited-scale training, consistent with observations in prior work (Xie et al., 2024). Instead, we leverage the tokenizer proposed in VILA-U (Wu et al., 2024b), which preserves image reconstruction quality while enhancing text-image alignment. Given an input image, the tokenizer compresses it by a factor of $16 \times 16$, flattens the resulting 2D features into a 1D token sequence, and passes it through a lightweight MLP before feeding the tokens into the language model.

**Transformer Backbone.** Motivated by our alignment analysis, UniFork adopts a Y-shaped Transformer architecture. Given a Transformer of $(M + N)$ total layers, the first $M$ layers are shared

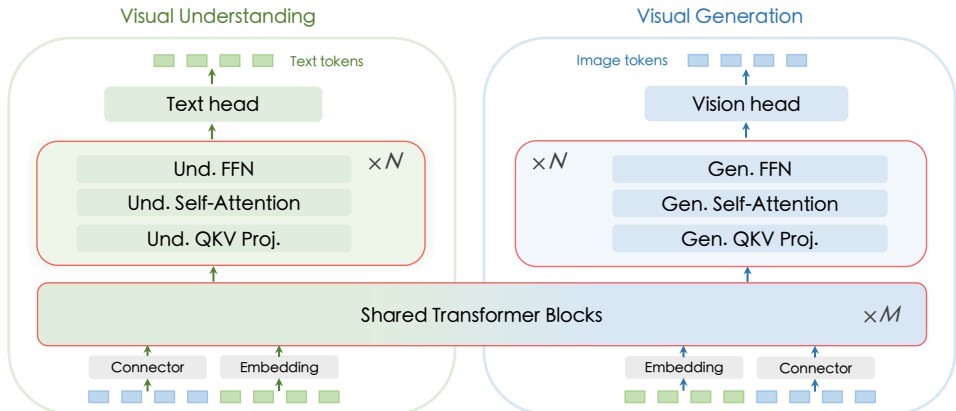

Figure 3: Overall framework of UniFork. UniFork adopts a Y-shaped Transformer backbone. The early layers are shared across both image generation and understanding tasks to facilitate joint semantic representation learning, while the later layers are split into task-specific branches to learn specialized representations. Und.: understanding. Gen.: generation. Proj.: projection.

across both tasks to support joint semantic representation learning. The remaining $N$ layers contains two task-specific branches: one dedicated to semantic reinforcement for image understanding, and the other focusing on spatial detail reconstruction for image generation. We initialize the entire backbone with weights from the Qwen2.5-0.5B LLM (Yang et al., 2025). Notably, when $N = 0$, UniFork reduces to the architecture of Emu3 (Wang et al., 2024) with full parameter sharing; when $M = 0$, it becomes structurally similar to the recently proposed Mixture-of-Transformers design in BAGEL (Deng et al., 2025).

**Generation Vision Head.** Since the image tokenizer (Wu et al., 2024b) uses the residual vector quantization method (Lee et al., 2022) to map each token into multiple discrete codes, we incorporate an image head to predict these codes. This head takes the output features from the last layer of LLM and generates codes for each token autoregressively.

### 3.3 TRAINING PIPELINE

As shown in Figure 4, the overall training process can be divided into three stages:

**Stage I: Visual Alignment Pretraining.** The objective of this stage is to align the visual representation with the pretrained LLM. Following prior works (Xie et al., 2024; Chen et al., 2025b), we first train the model on the ImageNet-1K (Deng et al., 2009) dataset to efficiently capture pixel-level dependencies. We formulate the learning task using paired images and textual descriptions, where class names are converted into natural language prompts using the OpenAI ImageNet templates (Radford et al., 2021). This data is used to train both image captioning and text-to-image generation. Subsequently, we perform training on the same two tasks with a mixture of 30 million samples from Laion-En (Schuhmann et al., 2022) and 10 million samples from COYO (Byeon et al., 2022). During this stage, the weights of the LLM are frozen, and we only train the randomly initialized visual connector and image head. The generation task follows the format `"<caption><image>"`, while the captioning task uses the format `"<image><caption>"`.

**Stage II: Joint Optimization.** This stage aims to enhance the model's overall ability in both image understanding and generation. We unfreeze the LLM and jointly optimize the backbone, visual connector, and image head. For the multitask pretraining, we use 32.5 million image-text pairs from JourneyDB (Sun et al., 2023a), SAM (Kirillov et al., 2023), Unsplash (Unsplash, 2020), and an internal dataset for generation, and a 16.5 million subset of InternVL-1.5 (Chen et al., 2024) pretraining data for understanding. We then perform instruction tuning. For generation, we sample a subset from the 32.5 million dataset and combine it with the BLIP3o-60k (Chen et al., 2025a) dataset, totaling 5 million samples. For understanding, we use a 3.8 million subset of the InternVL-1.5 SFT dataset. The format for generation task is: `"USER: <Input Message> ASSISTANT:`

Figure 4: Three-stage training pipeline for UniFork. The first stage focuses on aligning visual and textual modalities. The second stage performs joint training to enhance both image understanding and generation capabilities. In the third stage, task-specific parameters are alternately optimized using data from each task. Modules involved in training are highlighted in red.

`<Response>`". For the understanding task, we adopt the baseline SFT dialogue format (Yang et al., 2025).

**Stage III: Task-Specific Fine-Tuning.** An important advantage of the UniFork architecture is its flexibility in optimization. After joint training, we further refine task-specific performance through isolated fine-tuning. In this stage, only the task-specific layers are updated, while all shared components remain frozen. We reuse the instruction-tuning datasets from Stage II, and independently fine-tune the understanding and generation branches. This final stage allows the model to specialize in each task without introducing interference, effectively balancing shared semantic representation and task-specific optimization.

## 3.4 TRAINING OBJECTIVE

UniFork models both visual and textual tokens in an autoregressive manner. Therefore, we adopt the cross-entropy loss over both tasks, without introducing any task-specific loss weighting:

$$\mathcal{L}_{\text{total}} = -\sum_{i=1} \log P(\hat{x}_i = x_i \mid x_{<i}), \tag{1}$$

where $P$ denotes the probability distribution modeled by the UniFork network. $\hat{x}_i$ and $x_i$ represent the predicted and ground truth token respectively. For the image generation task, the loss is computed only over visual tokens. For the image understanding task, the loss is calculated solely on the response portion of the text tokens.

## 4 EXPERIMENTS

In this section, we present comprehensive experiments to evaluate the proposed UniFork structure. We begin with the experimental setup (Sec 4.1), followed by ablation studies that demonstrate the effectiveness of the Y-shaped architecture (Sec 4.2). Guided by these insights, we modestly scale the model and data, and compare UniFork against both expert models and recent unified models for image understanding and generation (Sec 4.3, 4.4). Finally, we analyze the modality alignment patterns of UniFork (Sec 4.5).

## 4.1 EXPERIMENT SETUP

**Implementation Details.** We initialize the Transformer backbone of UniFork using the Qwen2.5-0.5B LLM (Yang et al., 2025). The latter half of the Transformer layers are duplicated to construct two independent branches for image understanding and generation, respectively. The total number of parameters in the backbone is 1.21B, with 0.5B active parameters for understanding and 0.76B for generation. We adopt the tokenizer from VILA-U-256 (Wu et al., 2024b) to obtain text-aligned codes for each image. The tokenizer has a vocabulary size of $16,384$ with a compression factor of $16 \times 16$. Input images are resized to a resolution of $384 \times 384$. For image generation, we resize the shorter side to 384 and apply center crop on the longer side. For image understanding, the longer

side is resized to $384$, and the shorter side is padded with a background color (RGB: 127, 127, 127) to form a $384 \times 384$ input. Following previous works (Sun et al., 2024; Chen et al., 2025b), we employ the classifier-free guidance during image generation. Specifically, $10\%$ of input prompts are randomly dropped during training and replaced with a special padding token. During inference, the guidance scale is set to $4.0$ to balance fidelity and diversity. The training process is conducted on 16 Nvidia A100 GPUs, detailed configurations for each stage are summarized in the Appendix Table 7.

**Evaluation Benchmarks.** To evaluate the effectiveness of UniFork in both image understanding and generation, we compare it against expert models trained specifically for each task, as well as recent unified models. For image understanding, we conduct evaluations on five widely adopted benchmarks: MME-P (Fu et al., 2024), POPE (Li et al., 2023b), SEED-I (Li et al., 2023a), VQAv2 (Goyal et al., 2017), and GQA (Hudson & Manning, 2019). These benchmarks collectively assess various aspects of visual comprehension, including perception, reasoning, and grounding. For image generation, we use GenEval (Ghosh et al., 2023) and MJHQ-30K (Li et al., 2024) benchmarks. GenEval is an object-centric benchmark that assesses text-to-image alignment across six dimensions: "single objec", "two objects", "counting", "colors", "position", and "color attributes". While MJHQ-30K focuses on overall image quality and visual aesthetics. It uses the Fréchet Inception Distance (FID) metric to evaluate the similarity between generated images and a curated set of 30K high-quality reference images.

## 4.2 ABLATION STUDY

**Effectiveness of UniFork Structure.** To validate the effectiveness of the proposed UniFork architecture, we conduct a comparative study using four model variants, all initialized from the Qwen2.5-0.5B LLM (Yang et al., 2025). The variants include: (1) **Gen Expert**, trained exclusively on generation data; (2) **Und Expert**, trained exclusively on understanding data; (3) **Fully Shared LLM**, which uses a single Transformer backbone for both tasks, with a 0.07B vision head for image generation; and (4) **UniFork**, which shares the first half layers of the Transformer but adopts separate task-specific branches in the latter half. All models are trained on a subset of the data used in Stage I and Stage II, with the input image resolution set to $256 \times 256$. To ensure a fair comparison, we keep the number of activated parameters and training configurations consistent for each task.

As shown in Table 1, UniFork consistently outperforms the Fully Shared LLM on both image understanding and generation tasks, and achieves comparable or even better performance than the task-specific expert models. These results demonstrate that the Y-shaped Transformer architecture achieves a more effective trade-off between shared semantic learning and task-specific representation. By decoupling the later layers, UniFork reduces task interference and enables targeted feature refinement, leading to overall performance gains across both modalities.

Table 1: Ablation study to verify the effectiveness of UniFork architecture.

| Model | Und. Evaluation | | | Gen. Evaluation | |
|---|---|---|---|---|---|
| | MME-P↑ | VQAv2↑ | SEED-I↑ | GenEval↑ | MJHQ↓ |
| Gen Expert | - | - | - | 0.36 | 15.1 |
| Und Expert | 1177 | 69.7 | 56.6 | - | - |
| Fully Shared LLM | 1203 | 69.6 | 53.9 | 0.28 | 17.2 |
| UniFork (ablation) | 1229 | 69.9 | 55.1 | 0.33 | 16.3 |

**The Ratio between Shared and Task-Specific Layers.** To determine the optimal proportion of shared layers, we initialize three models from the Qwen2.5-0.5B LLM (Yang et al., 2025), each configured with a different number of shared transformer layers. All models are trained on the same dataset with a fixed image resolution of $256 \times 256$ to ensure a fair comparison.

As shown in Table 2, sharing 12 layers—corresponding to half of the transformer depth—achieves the best performance on both understanding and generation tasks. This suggests that moderate parameter sharing enhances cross-task representation learning and improves both understanding and generation performance. However, in deeper layers, the objectives of understanding and generation progressively diverge; excessive sharing in these layers introduces task interference, ultimately degrading performance.

Table 2: Ablation study on the ratio between shared and task-specific layers.

| Number of shared layers ($M$) | MME↑ | POPE↑ | GenEval↑ |
|:---:|:---:|:---:|:---:|
| 8 | 1368 | 82.9 | 0.28 |
| 12 | 1402 | 84.5 | 0.29 |
| 16 | 1322 | 84.0 | 0.25 |

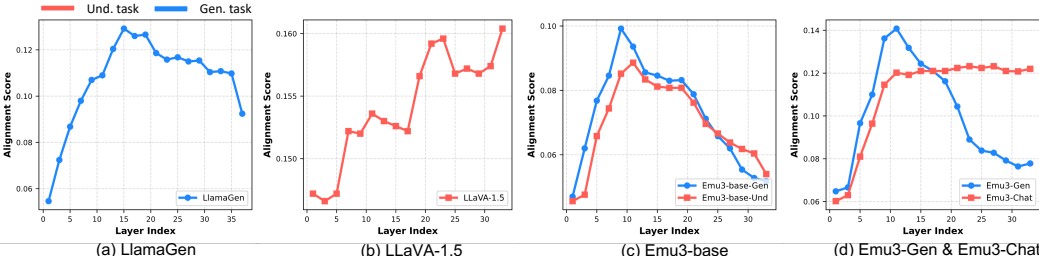

(a) LlamaGen     (b) LLaVA-1.5     (c) Emu3-base     (d) Emu3-Gen & Emu3-Chat

Figure 5: Modality alignment analysis on MJHQ-30K. The observed alignment patterns on this dataset are consistent with those reported in Section 3.1.

**Modality Alignment Analysis on More Datasets.** In Section 3.1, we analyzed modality alignment patterns using the GenEval (Ghosh et al., 2023) benchmark. Its prompts are relatively short and primarily focus on object-level descriptions. To avoid potential dataset-specific biases, we extend our alignment analysis to longer prompts with more emphasis on holistic scene descriptions and stylistic attributes.

We randomly sample 500 prompts from MJHQ-30K (Li et al., 2024), with the average prompt length increasing from 7.3 words in GenEval (Ghosh et al., 2023) to 32.9 words. As shown in Figure 5, the observed trends remain consistent with those in Figure 2. Specifically, the alignment score for generation still exhibits a rise-then-fall pattern across Transformer layers, while the alignment for understanding continues to increase monotonically. These results further support our earlier findings: fully sharing a Transformer backbone may lead to representational compromise between the two tasks. This highlights the necessity of the UniFork architecture to better accommodate the divergent alignment behaviors of image generation and understanding within a unified framework.

### 4.3 IMAGE UNDERSTANDING EVALUATION

As shown in Table 3, UniFork delivers strong performance across all image understanding benchmarks, despite using only 0.5B active parameters for this task. Compared to the recent unified model Show-o (Xie et al., 2024) (1.3B), UniFork achieves relative gains of $10\%$ on MME-P and $7.3\%$ on POPE. Notably, UniFork remains competitive even against larger understanding-only models, such as MobileVLM (2.7B) (Chu et al., 2023), IDEFICS-9B (Laurençon et al., 2023), and LLaVA (7B) (Liu et al., 2023). It matches MobileVLM on POPE ($85.8$ vs. $84.9$), and outperforms IDEFICS-9B on SEEDv1 ($55.2$ vs. $45.0$). We further provide some qualitative results in the Appendix Figure 8. These results highlight the effectiveness of our Y-shaped architecture. It helps reduce task interference and enables UniFork to perform well even with a limited parameter budget.

### 4.4 IMAGE GENERATION EVALUATION

As shown in Table 4, UniFork achieves an overall $46\%$ accuracy on GenEval, representing a $39\%$ improvement over the ablation variant with smaller parameter scale. Notably, UniFork outperforms prior unified models with similar or larger sizes, such as LWM (Liu et al., 2024a) and Chameleon (Team, 2024), and also surpasses several generation-only baselines, including LDM (Rombach et al., 2022a) and LlamaGen (Sun et al., 2024), across most categories. On MJHQ-30K (Table 5), UniFork achieves a FID score of 10.6, marking a $35\%$ improvement over its smaller variant. This FID also surpasses previous unified models such as Show-o (15.18) (Xie et al., 2024) and LWM (17.77), despite UniFork using significantly fewer parameters (0.76B vs. 7B+). We attribute these gains to the structural insights from our ablation study, which demonstrated that task-

Table 3: Evaluation results on multimodal understanding benchmarks. "# LLM A-Params" denotes the number of activated parameters of Transformer backbone during inference.

| Type | Model | # LLM A-Params | MME-P↑ | POPE↑ | SEEDv1↑ | VQAv2↑ | GQA↑ |
|---|---|---|---|---|---|---|---|
| *Und. Only* | MobileVLM (Chu et al., 2023) | 1.4B | 1196.2 | 84.5 | - | - | 56.1 |
| | MobileVLM (Chu et al., 2023) | 2.7B | 1288.9 | 84.9 | - | - | 59.0 |
| | LLaVA (Liu et al., 2023) | 7B | 809.6 | 76.3 | 33.5 | - | - |
| | IDEFICS-9B (Laurençon et al., 2023) | 8B | 1177.3 | 81.9 | 45.0 | - | 38.4 |
| | InstructBLIP (Dai et al., 2023) | 7B | - | - | 53.4 | - | 49.5 |
| | LLaVA-v1.5-Phi-1.5 (Xie et al., 2024) | 1.3B | 1128.0 | 84.1 | - | 75.3 | - |
| *Und. and Gen.* | Emu (Sun et al., 2023b) | 13B | - | - | - | 52.0 | - |
| | Gemini-Nano-1 (Team et al., 2023) | 1.8B | - | - | - | 62.7 | - |
| | LaVIT (Jin et al., 2023) | 7B | - | - | - | 66.0 | 46.8 |
| | LWM (Liu et al., 2024a) | 7B | - | 75.2 | - | - | 44.8 |
| | NExT-GPT (Wu et al., 2024a) | 13B | - | - | - | 66.7 | - |
| | Chameleon (Team, 2024) | 7B | - | - | 30.5 | 66.0 | - |
| | Show-o (Xie et al., 2024) | 1.3B | 1097.2 | 80.0 | - | 69.4 | 58.0 |
| | D-Dit (Li et al., 2025b) | 2B | 1124.7 | 84.0 | - | - | 59.2 |
| | UniFork (main) | 0.5B | 1208.0 | 85.8 | 55.2 | 70.0 | 55.1 |

specific branches are essential for resolving modality alignment conflicts in unified training. We further provide some qualitative results in Figure 1 and Appendix Figure 7.

By modestly scaling the model from 0.57B to 0.76B active parameters, we unlock substantial performance improvements without requiring architectural changes. We expect further improvements with better tokenizers, larger parameters and higher-quality data.

Table 4: Image generation results on GenEval dataset. "# LLM A-Params" denotes the number of activated parameters of Transformer backbone during inference. Obj.: Object. Attri.: Attribution.

| Type | Model | # LLM A-Params | Single Obj. | Two Obj. | Counting | Colors | Position | Color Attri. | Overall↑ |
|---|---|---|---|---|---|---|---|---|---|
| *Gen. Only* | LDM (Rombach et al., 2022a) | 1.4B | 0.92 | 0.29 | 0.23 | 0.70 | 0.02 | 0.05 | 0.37 |
| | SDv1.5 (Rombach et al., 2022a) | 0.9B | 0.97 | 0.38 | 0.35 | 0.76 | 0.04 | 0.06 | 0.43 |
| | LlamaGen (Sun et al., 2024) | 0.8B | 0.71 | 0.34 | 0.21 | 0.58 | 0.07 | 0.04 | 0.32 |
| | SEED-X (Ge et al., 2024b) | 17B | 0.97 | 0.58 | 0.26 | 0.8 | 0.19 | 0.14 | 0.49 |
| | LWM (Liu et al., 2024a) | 7B | 0.93 | 0.41 | 0.46 | 0.79 | 0.09 | 0.15 | 0.47 |
| *Und. and Gen.* | Chameleon (Team, 2024) | 34B | - | - | - | - | - | - | 0.39 |
| | UniFork (ablation) | 0.57B | 0.83 | 0.24 | 0.16 | 0.62 | 0.06 | 0.62 | 0.33 |
| | UniFork (main) | 0.76B | 0.95 | 0.50 | 0.22 | 0.73 | 0.19 | 0.17 | 0.46 (↑39%) |

Table 5: Image generation results on MJHQ-30K dataset. "# LLM A-Params" denotes the number of activated parameters of Transformer backbone during inference.

| Type | Model | # LLM A-Params | MJHQ↓ |
|---|---|---|---|
| *Gen. Only* | LDM (Rombach et al., 2022a) | 1.4B | - |
| | SDv1.5 (Rombach et al., 2022a) | 0.9B | - |
| | LlamaGen (Sun et al., 2024) | 0.8B | - |
| *Und. and Gen.* | LWM (Liu et al., 2024a) | 7B | 17.77 |
| | VILA-U-256 (Wu et al., 2024b) | 7B | 12.81 |
| | Show-o (Xie et al., 2024) | 1.3B | 15.18 |
| | UniFork (ablation) | 0.57B | 16.3 |
| | UniFork (main) | 0.76B | 10.6 (↑35%) |

## 4.5 MODALITY ALIGNMENT ANALYSIS

Following the methodology introduced in Section 3.1, we further visualize the modality alignment patterns of UniFork for both image understanding and generation tasks. As shown in Figure 6, the alignment score for the understanding task increases steadily with network depth, while that for the generation task follows a rise-then-fall trend. These trends are consistent with those observed in the expert models, satisfying the distinct representational needs of the two tasks.

This result provides additional evidence for the effectiveness of the UniFork architecture. By decoupling the later layers of the Transformer backbone and assigning task-specific parameters, the model

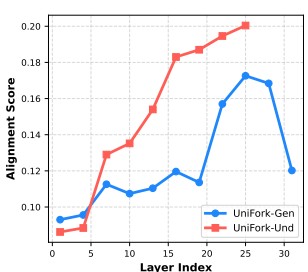

Figure 6: Modality alignment analysis for UniFork.

can reconcile the divergent requirements of generation and understanding within a unified framework, without forcing a compromise in representation quality.

### 4.6 APPLICATION AND SCALABILITY

The core idea of UniFork—learning shared semantic representations in shallow layers and then applying task specialization in deep layers—is not restricted to a particular model structure. To evaluate its scalability, we apply our framework to VILA-U (Wu et al., 2024b) with 7B parameters. Concretely, we duplicate the latter half of the VILA-U Transformer layers to construct an understanding-specific branch, yielding **VILA-U-UniFork**. During fine-tuning, only these understanding-specific layers are updated using Stage-III data from the UniFork pipeline, while all generation-related parameters remain frozen to keep its original generative ability.

As shown in Table 6, VILA-U-UniFork achieves consistent improvements across multiple understanding benchmarks. This demonstrates that UniFork can be seamlessly scaled to larger models without structural modification for each task, further validating its scalability, generality, and effectiveness.

Table 6: Scalability validation of UniFork architecture.

| Model | # LLM A-Params | MME-P↑ | POPE↑ | SEEDv1↑ | VQAv2↑ |
|---|---|---|---|---|---|
| VILA-U | 7B | 1341 | 84.9 | 61.7 | 74.0 |
| VILA-U-UniFork | 7B | 1358 | 85.3 | 63.7 | 74.3 |

## 5 CONCLUSION

In this paper, we analyzed modality alignment patterns in expert models and NTP-based unified models for image generation and understanding. We found that fully sharing a Transformer backbone may lead to task interference. Inspired by this finding, we proposed UniFork that shares early layers and decouples the later ones for task-specific learning. Ablation studies validate the effectiveness of this design. With modest scaling, UniFork achieves strong performance on both tasks, demonstrating its potential as a baseline for future unified multimodal models.

## 6 LLM USAGE

We note that LLMs were used solely as assistive tools to polish the writing and coding. They were not involved in research ideation, experimental design, or analysis.

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

## A   TRAINING CONFIGURATIONS

Detailed configurations for each training stage are listed in Table 7.

Table 7: Detailed training configurations of UniFork.

| Configuration | Stage 1 | Stage 2 | | Stage 3 | |
|---|---|---|---|---|---|
| | | Pretrain | SFT | Gen. task | Und. task |
| Learning rate | $1e^{-4}$ | $5e^{-5}$ | $5e^{-5}$ | $3e^{-5}$ | $2e^{-5}$ |
| LR scheduler | Constant | Constant | Constant | Constant | Cosine |
| Warm-up steps | 0.03 | 0 | 0 | 0 | 0.03 |
| Training steps | 125K | 170K | 23K | 17K | 15K |
| Global batch size | 384 | 384 | 384 | 384 | 256 |
| Weight decay | | | 0.0 | | |
| Optimizer | | | AdamW | | |
| Context Length | | | 1350 | | |
| Numerical precision | | | `bfloat16` | | |

## B   QUALITATIVE RESULTS

We provide some qualitative results for image generation and understanding in Figure 7 and Figure 8, respectively.

| SDv1.5 | LlamaGen | UniFork (ours) | SDv1.5 | LlamaGen | UniFork (ours) |
|---|---|---|---|---|---|

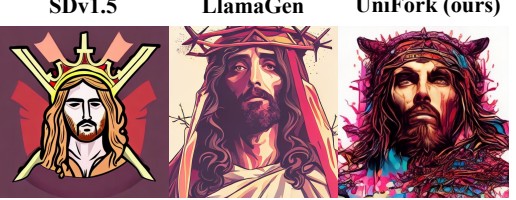

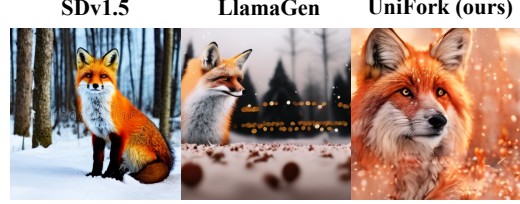

"jesus wearing crown of thorns, brutal, vivid colors, lots of details, sticker, horror."

"closeup portrait of a red fox in swedish forrest at sunset in winter, light snowfall life style stock image popular no text prompt trend."

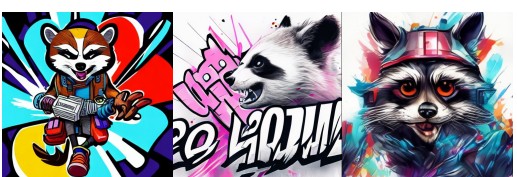

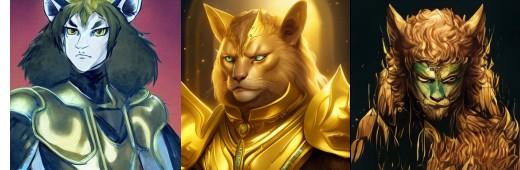

"Rocket Raccoon in style of new school tattoo, white background, bright colors, 3d graffiti effect, cartoon look, exaggerated facial features, ultra HD, 8K."

"lyrian, handsome human man with cat ears and lion-like face, cat-like ears, Golden ceremonial armor, strong green eyes, strong sense of pride and compassion."

Figure 7: Qualitative results on the text-to-image generation task. We compare image samples generated by SDv1.5, LlamaGen, and UniFork, with respective resolutions of $512 \times 512$, $512 \times 512$, and $384 \times 384$.

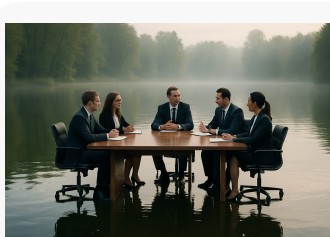

Do you think the image is unusual or not? Why?

**UniFork (0.5B)**: The image is unusual because it features a group of people sitting around a table in the middle of a **lake**, rather than in a typical office setting. This is not a common scenario, as people typically work in office environments. The presence of a lake and the unusual setting of the table and chairs make the scene quite unique and unexpected.

**Chameleon (7B)**: The image is unusual because it features five people sitting at a table in a **park**, discussing something. The presence of five people and the nature of their conversation suggests that this may be a meeting or a group discussion. The fact that they are all dressed professionally and are sitting at a table with papers and pens suggests that they may be discussing work-related matters. The setting in a park also adds a unique and unexpected twist to the scene, as it is not typically associated with formal business discussions. Overall, the image is unusual because it presents a scene that is not commonly found in everyday life.

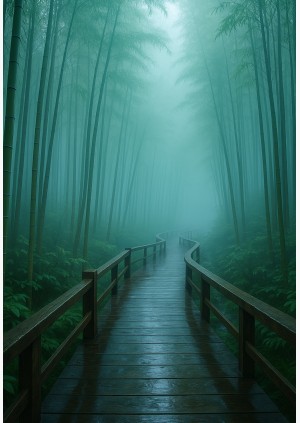

What are the things I should be cautious about when I visit here?

**UniFork (0.5B)**: When visiting this foggy, misty forest with a wooden bridge, you should be cautious about several things. Firstly, be aware of the potential for slippery surfaces due to the wet and damp conditions, which can make walking on the bridge slippery. It's essential to wear appropriate footwear with good traction to avoid slipping. Secondly, be mindful of the surrounding wildlife, as they can be unpredictable and may be startled by the fog. Look out for any signs of predators or snakes in the area. Lastly, be cautious of any obstacles or hazards on the bridge, such as rocks or fallen branches, that could cause accidents. By being cautious and attentive, you can ensure a safe and enjoyable experience in this serene and misty forest setting.

**Chameleon (7B)**: It is important to be aware of your surroundings when hiking and ensure that you are prepared for any challenges that may arise. Before embarking on your journey, be sure to check the weather forecast and dress appropriately. Always bring plenty of water and snacks to keep you fueled throughout your hike. It is also crucial to let someone know your itinerary and estimated return time, in case of an emergency. Additionally, be mindful of your trail etiquette and respect the environment by leaving no trace. By following these guidelines, you can enjoy your hike while minimizing the risk of injury or encountering any challenges.

Figure 8: Qualitative results on the image understanding task. The key points in the answers are highlighted in red.

