# OpenReview forum: "UniFork: Exploring Modality Alignment for Unified Multimodal Understanding and Generation"
_ICLR.cc/2026/Conference — Submitted to ICLR 2026_

### Official Review · Reviewer_P6ZU · 2025-10-16

**Soundness:** 3
**Presentation:** 3
**Contribution:** 2
**Rating:** 6
**Confidence:** 4

**Summary:**

Empirical characterization of divergent modality-alignment patterns for understanding vs generation and evidence of representational compromise in fully shared NTP models.

UniFork Y-architecture that shares early layers and decouples late layers into task-specific branches while keeping a simple AR/NTP interface.

Simple, effective training recipe (three stages, no loss reweighting) that allows independent branch fine-tuning without delicate data balancing.

**Strengths:**

The paper’s exploration of modality alignment patterns in understanding versus generation tasks fills a gap in prior research by investigating how these tasks differ at a deep level. The observation that understanding benefits from progressively increasing alignment, while generation requires an initial rise and later drop, is a novel and insightful contribution to the multimodal field.

The presentation of the Y-shaped Transformer architecture is clear, and the distinction between the shared early layers and task-specific branches is well-articulated. The decision to share the first half of the layers and split the latter half for task-specific learning is explained in an understandable manner, making the technical concept more accessible without oversimplifying it.

**Weaknesses:**

The paper correctly notes that when M=0, UniFork is structurally similar to Mixture-of-Transformers designs like BAGEL, and its dual-pathway approach is conceptually related to the Janus series. However, the results tables lack a direct, quantitative comparison with these specific models under similar training budgets. This omission makes it difficult for the reader to gauge whether the Y-shaped architecture provides a tangible advantage over other state-of-the-art parameter-efficient unification strategies.

Actionable Improvement: Include direct comparisons with BAGEL and/or a Janus-series model in Tables 2, 3, and 4. This would precisely delineate UniFork's contribution within the existing landscape of unified model architectures.

The performance jump from the 0.57B to 0.76B ablation variant is promising, but it's a single data point. A critical unanswered question is how the UniFork architecture behaves at different scales (e.g., with a 3B or 7B backbone). Does the optimal ratio of shared-to-task-specific layers (M and N) change with model size?

Actionable Improvement: Conduct a basic scaling study. For instance, implement UniFork on two different backbone sizes (e.g., 0.5B and 2B) and show that the architecture's benefits are consistent. This would significantly bolster the claim that it is a general-purpose design.


The generative performance is assessed primarily on GenEval (text-image alignment) and FID (general quality). This misses important dimensions like compositional reasoning, complex scene rendering, and aesthetic quality.

Actionable Improvement: Supplement the evaluation with benchmarks like T2I-CompBench， WISE for compositionality and, ideally, a human evaluation study to assess image realism and prompt adherence on more complex, creative prompts.

The "Where to Fork" Ablation is Missing: The analysis reveals that the alignment trends diverge, but the choice to decouple specifically in the later layers is a design decision. The paper lacks an ablation study that justifies this specific choice. Is forking in the final third of the network truly optimal? What is the performance impact of forking earlier (e.g., after the first third) or later?

Actionable Improvement: Introduce a critical ablation experiment that varies the forking point M within the Transformer stack while holding total depth constant. A plot showing task performance versus forking depth would provide direct, empirical evidence that the chosen configuration is optimal, transforming a design choice into a data-driven conclusion.

**Questions:**

see weakness

---

> ### Author Response · Authors · 2025-11-21
> **Official Comment by Authors**
>
> We sincerely thank the reviewer for the positive assessment and strong recognition of our method. Below is our detailed response to clarify the points and answering the concerns you raised.
>
> **W2: Scalability validation of UniFork architecture**
>
> We added the experimental results in the table below and Table 6 of the revised paper.
>
> | Model | # LLM A-Params | MME-P ↑ | POPE ↑ | SEEDv1 ↑ | VQAv2 ↑ |
> | --- | --- | --- | --- | --- | --- |
> | VILA-U | 7B | 1341 | 84.9 | 61.7 | 74.0 |
> | VILA-U-UniFork | 7B | 1358 | 85.3 | 63.7 | 74.3 |
>
> We apply our framework to VILA-U with 7B parameters. Concretely, we duplicate the latter half of the VILA-U Transformer layers to construct an understanding-specific branch, yielding VILA-U-UniFork. During fine-tuning, only these understanding-specific layers are updated using Stage-III data from the UniFork pipeline, while all generation-related parameters remain frozen to keep its original generative ability.
>
> As shown in the table, VILA-U-UniFork achieves consistent improvements across multiple understanding benchmarks. This demonstrates that UniFork can be seamlessly scaled to larger models without structural modification for each task, further validating its scalability, generality, and effectiveness.
>
> **W4: Ablation on the number of layers for shared and unshared layers.**
>
> We have added the ablation results in the table below and in Table 2 of the revised paper.
>
> | Number of shared layers ($M$) | MME ↑ | POPE ↑ | GenEval ↑ |
> | --- | --- | --- | --- |
> | 8 | 1368 | 82.9 | 0.28 |
> | 12 | 1402 | 84.5 | 0.29 |
> | 16 | 1322 | 84.0 | 0.25 |
>
> We experiment with different numbers of shared layers while keeping the training settings identical. Sharing 12 layers (half of the transformer depth) achieves the best performance across both understanding and generation tasks. This suggests that a moderate level of sharing facilitates cross-task representation learning. However, in deeper layers, the objectives of understanding and generation progressively diverge; excessive sharing in these later stages introduces task interference, ultimately degrading performance.
>
> **W1 & W3: Limited baselines and generation benchmarks**
>
> Thank you for raising this point. While we acknowledge that including more baselines and benchmarks would be valuable, achieving SOTA performance against these systems requires substantial compute and high-quality data. As our work primarily focuses on the architectural insight, we aimed to validate the design under realistic research-scale resources rather than pursue an exhaustively benchmarked SOTA unified model.
>
> To further support your concern, we have added scalability validation in Table 6, complementing the existing architectural ablations, fork-depth analysis and benchmark evaluation. We believe that this evaluation suite collectively demonstrates the effectiveness and generalizability of the proposed Y-shaped architecture.

---

### Official Review · Reviewer_5cnP · 2025-10-29

**Soundness:** 2
**Presentation:** 3
**Contribution:** 2
**Rating:** 2
**Confidence:** 5

**Summary:**

The paper introduces UniFork, a novel Y-shaped architecture that shares the shallow layers for cross-task representation learning for unified model. The authors diagnose that image understanding and image generation tasks demand different cross-modal feature alignment behaviours in transformer layers. Experiments demonstrate performance gains relative to fully shared architectures, achieving similar or better results compared to task-specific expert models.

**Strengths:**

1. Insightful analysis of modality alignment patterns: The paper provides a systematic investigation into the alignment dynamics between text and image representations across different architectures and tasks (understanding vs. generation). By empirically identifying distinct alignment patterns and connecting them to architectural design choices, the work offers valuable conceptual insights for building more principled and efficient unified multimodal models.
2. Well-documented training and implementation details: The paper reports detailed information on training strategy, data composition ratios, and optimization settings, which facilitates reproducibility.

**Weaknesses:**

1. Outdated and weak baselines: The compared baselines (e.g., MobileVLM, Emu, LaVIT, LDM, LWM) are mostly early-generation unified or multimodal models whose performance and architecture are now considerably behind state-of-the-art models such as Emu3, Bagel, or Janus-Pro. Even when compared to these relatively weak baselines, UniFork shows only marginal or inconsistent improvements across several benchmarks. This weakens the empirical strength of the claimed performance gains and raises questions about whether the proposed method would still hold advantages under stronger baselines.
2. Lack of ablation or sensitivity analysis on branch duplication and split ratio: The proposed UniFork architecture duplicates the latter half of the Transformer backbone (based on Qwen2.5) to construct two independent branches for understanding and generation. However, there is no thorough analysis or comparison regarding this specific design choice—e.g., why the fork point is placed in the latter half, and how sensitive the results are to the split ratio between shared and task-specific layers.
As described in §3.2 (“Given a Transformer of (M + N) total layers”), the determination of M and N (the numbers of shared and branch-specific layers, respectively) seems largely heuristic. A systematic study or empirical justification of these hyperparameters would substantially strengthen the paper’s credibility and generality.

**Questions:**

1. Choice of fork depth (M, N): In §3.2 the authors mention that the Transformer backbone is divided into M shared and N task-specific layers (“Given a Transformer of (M + N) total layers”). How was the specific fork point determined in practice?
  - Was it based on alignment measurements, empirical tuning, or a fixed heuristic?
  - Have you explored sensitivity analysis on different M/N splits (e.g., forking earlier vs. later), and how does this affect performance or modality alignment?
2. Scalability and future experiments: The current experiments are conducted on relatively moderate-scale models. Do the authors plan to extend UniFork to larger-scale settings to further validate the scalability and general effectiveness of the proposed architecture?
It would be valuable to know whether the observed alignment patterns and performance trends remain consistent at higher scales.

---

> ### Author Response · Authors · 2025-11-21
> **Official Comment by Authors**
>
> Thank you so much for the valuable feedback and suggestion.
>
> **W2 & Q1: Ablation or sensitivity analysis on branch duplication and split ratio.**
>
> We have added the ablation results in the table below and in Table 2 of the revised paper.
>
> | Number of shared layers ($M$) | MME-S ↑ | POPE ↑ | GenEval ↑ |
> | --- | --- | --- | --- |
> | 8 | 1368 | 82.9 | 0.28 |
> | 12 | 1402 | 84.5 | 0.29 |
> | 16 | 1322 | 84.0 | 0.25 |
>
> We experiment with different numbers of shared layers while keeping the training settings identical. Sharing 12 layers (half of the transformer depth) achieves the best performance across both understanding and generation tasks. This suggests that a moderate level of sharing facilitates cross-task representation learning. However, in deeper layers, the objectives of understanding and generation progressively diverge; excessive sharing in these later stages introduces task interference, ultimately degrading performance.
>
> **Additional experiments are in progress**
>
> We also carefully reviewed your comments and started the additional experiments. We are accelerating the process as much as possible within our compute budget. We will update the revised version and discuss with you as soon as the new results are ready.

---

> > ### Comment · Reviewer_5cnP · 2025-11-27
> >
> > Thank you for your reply.
> > My concerns have been largely resolved, so I will raise my rating.

---

> > > ### Author Response · Authors · 2025-11-27
> > > **Official Comment by Authors**
> > >
> > > Thank you very much for your follow-up and for raising the rating — we truly appreciate your recognition. Below we address your remaining concerns in detail:
> > >
> > > **W1: Limited baselines**
> > >
> > > Thank you for raising this point. While we acknowledge that including more baselines would be valuable, achieving SOTA performance against these systems requires substantial compute and high-quality data. As our work primarily focuses on the architectural insight, we aimed to validate the design under realistic research-scale resources rather than pursue an exhaustively benchmarked SOTA unified model.
> > >
> > > To further support your concern, we have added scalability validation in Table 6, complementing the existing architectural ablations, fork-depth analysis and benchmark evaluation. We believe that this evaluation suite collectively demonstrates the effectiveness and generalizability of the proposed Y-shaped architecture.
> > >
> > > **Q2: Scalability validation of UniFork architecture**
> > >
> > > We added the experimental results in the table below and Table 6 of the revised paper.
> > >
> > > | Model | # LLM A-Params | MME-P ↑ | POPE ↑ | SEEDv1 ↑ | VQAv2 ↑ |
> > > | --- | --- | --- | --- | --- | --- |
> > > | VILA-U | 7B | 1341 | 84.9 | 61.7 | 74.0 |
> > > | VILA-U-UniFork | 7B | 1358 | 85.3 | 63.7 | 74.3 |
> > >
> > > We apply our framework to VILA-U with 7B parameters. Concretely, we duplicate the latter half of the VILA-U Transformer layers to construct an understanding-specific branch, yielding VILA-U-UniFork. During fine-tuning, only these understanding-specific layers are updated using Stage-III data from the UniFork pipeline, while all generation-related parameters remain frozen to keep its original generative ability.
> > >
> > > As shown in the table, VILA-U-UniFork achieves consistent improvements across multiple understanding benchmarks. This demonstrates that UniFork can be seamlessly scaled to larger models without structural modification for each task, further validating its scalability, generality, and effectiveness.

---

### Official Review · Reviewer_k4Sk · 2025-10-30

**Soundness:** 3
**Presentation:** 3
**Contribution:** 3
**Rating:** 4
**Confidence:** 4

**Summary:**

The paper analyzes the variation curves of modality alignment scores in the intermediate layers of understanding models and generative models from the perspective of modality alignment. Based on the differences in the score variation curves between understanding and generative models, the paper proposes UniFork, a novel Y-shaped architecture that shares shallow layers for cross-task representation learning while employing task-specific branches in deeper layers to avoid task interference. The proposed model shows a significant improvement compared to baseline models.

**Strengths:**

1. This paper innovatively analyzes unified generative large models from a modality alignment perspective. Based on this analysis, the proposed Y-shape structure is experimentally tested and demonstrates good performance
2. The paper's presentation is good; the figures clearly convey the conclusions and experimental results of the paper.

**Weaknesses:**

1. The scale of the experiment is insufficient. The experimental model in the paper is only 0.5B~0.76B in size, which is too small compared to other existing unified understanding-generation models. I personally believe that it is necessary to further expand the model scale and verify the effectiveness of the method with a larger number of parameters.
2. The other methods compared in the paper are somewhat outdated. For example, widely accepted and published unified large model papers such as show-o2, tokenflow, and unitok were not compared or discussed
3. The paper did not conduct experiments on the number of layers for shared and unshared layers.

**Questions:**

Please see weakness
1.  Regarding the curve of intermediate layer alignment scores in generative models, it shows a distribution of first rising and then falling. However, the author only conducted comparative experiments on generative models using a pixel tokenizer and neglected the unified tokenizer. If semantic information is introduced in the tokenizer, would such an alignment score curve still hold?

---

> ### Author Response · Authors · 2025-11-21
> **Official Comment by Authors**
>
> We sincerely appreciate your constructive and thorough comments. Below is our detailed response to clarify the points and answering the concerns you raised.
>
> **W1: Scalability validation of UniFork architecture**
>
> We added the experimental results in the table below and Table 6 of the revised paper.
>
> | Model | # LLM A-Params | MME-P ↑ | POPE ↑ | SEEDv1 ↑ | VQAv2 ↑ |
> | --- | --- | --- | --- | --- | --- |
> | VILA-U | 7B | 1341 | 84.9 | 61.7 | 74.0 |
> | VILA-U-UniFork | 7B | 1358 | 85.3 | 63.7 | 74.3 |
>
> We apply our framework to VILA-U with 7B parameters. Concretely, we duplicate the latter half of the VILA-U Transformer layers to construct an understanding-specific branch, yielding VILA-U-UniFork. During fine-tuning, only these understanding-specific layers are updated using Stage-III data from the UniFork pipeline, while all generation-related parameters remain frozen to keep its original generative ability.
>
> As shown in the table, VILA-U-UniFork achieves consistent improvements across multiple understanding benchmarks. This demonstrates that UniFork can be seamlessly scaled to larger models without structural modification for each task, further validating its scalability, generality, and effectiveness.
>
> **W2: Limited baselines**
>
> Thank you for raising this point. While we acknowledge that including more baselines would be valuable, achieving SOTA performance against these systems requires substantial compute and high-quality data. As our work primarily focuses on the architectural insight, we aimed to validate the design under realistic research-scale resources rather than pursue an exhaustively benchmarked SOTA unified model.
>
> To further support your concern, we have added scalability validation in Table 6, complementing the existing architectural ablations, fork-depth analysis and benchmark evaluation. We believe that this evaluation suite collectively demonstrates the effectiveness and generalizability of the proposed Y-shaped architecture.
>
> **W3: Ablation on the number of shared vs. task-specific layers**
>
> We have added the ablation results in the table below and in Table 2 of the revised paper.
>
> | Number of shared layers ($M$) | MME ↑ | POPE ↑ | GenEval ↑ |
> | --- | --- | --- | --- |
> | 8 | 1368 | 82.9 | 0.28 |
> | 12 | 1402 | 84.5 | 0.29 |
> | 16 | 1322 | 84.0 | 0.25 |
>
> We experiment with different numbers of shared layers while keeping the training settings identical. Sharing 12 layers (half of the transformer depth) achieves the best performance across both understanding and generation tasks. This suggests that a moderate level of sharing facilitates cross-task representation learning. However, in deeper layers, the objectives of understanding and generation progressively diverge; excessive sharing in these later stages introduces task interference, ultimately degrading performance.
>
> **Q1: The comparison only uses pixel tokenizers; would the alignment curve still hold with unified tokenizers incorporating semantics?**
>
> Thank you for raising this point. In Figure 1, our analysis of the alignment pattern for the generation task is conducted on Emu3 and LlamaGen, which indeed use pixel tokenizers.
>
> In our UniFork experiments, however, we adopt the VILA-U unified tokenizer, which is jointly trained with semantic distillation and image reconstruction. As shown in Figure 6, the generation task under UniFork exhibits the same raise-and-fall trend in alignment scores. This demonstrates that the observed alignment behavior is consistent across both pixel-based tokenizers and semantically enriched unified tokenizers.

---

### Official Review · Reviewer_a4kZ · 2025-10-31

**Soundness:** 2
**Presentation:** 2
**Contribution:** 2
**Rating:** 4
**Confidence:** 2

**Summary:**

This paper introduces UniFork, a Y-shaped Transformer model designed to handle both image understanding and image generation in one model. It works by sharing early layers for general learning and splitting into task-specific branches later. UniFork achieves strong results on both understanding and generation tasks, outperforming several understanding-only and unified models, despite of being much smaller in size.

**Strengths:**

- The paper is mostly well written and structured. The core idea—the conflict in alignment patterns and the Y-shaped solution—is easy to follow.
- The alignment pattern analysis is insightful and seems new.
- On the chosen evaluation benchmarks, UniFork has shown stronger performance than the ablated instances and prior models.

**Weaknesses:**

- Limited Understanding Benchmarks. The understanding benchmarks tested in this paper is quite outdated (e.g., VQAv2 and GQA). For visual perception, consider including benchmarks like MMBench, BLINK, CVBench, MM-VET, MMVP.
- Limited baseline comparison, although Janus Pro and Bagel are cited, they are being compared against UniFork.
- The paper fixes the split point between shared and task-specific layers, but doesn’t show what happens when you change how many layers are shared (e.g., early split vs. late split). This would test how sensitive UniFork is to the choice of fork depth and whether the proposed setting is optimal.

Minor Presentation Issues
- Figure 7 caption: clouseup -> closeup
- the GenEval benchmark is spelled “GenEval” in some places and “Geneval” in others

**Questions:**

In addition to the weakness above, please find additional questions below.

- Is symmetric branch really necessary? As it is task-specific head anyways, would asymmetric branches (e.g., even more branches for generation) work just as well or better?

---

> ### Author Response · Authors · 2025-11-21
> **Official Comment by Authors**
>
> We sincerely thank the reviewer for the constructive and helpful feedback. Below is our detailed response to clarify the points and answering the concerns you raised.
>
> **W1 & W2: Limited understanding benchmarks and baselines:**
>
> Thank you for raising this point. While we acknowledge that expanding understanding benchmarks and including more baselines would be valuable, such scaling requires additional domain-aligned data and compute to ensure fair comparison and meaningful conclusions. As our work primarily focuses on the architectural insight, we aimed to validate the design under realistic research-scale resources rather than pursue an exhaustively benchmarked SOTA unified model.
>
> To further support the reviewer’s concern, we have added scalability validation in Table 6, complementing the existing architectural ablations, fork-depth analysis and benchmark evaluation. We believe that this evaluation suite collectively demonstrates the effectiveness and generalizability of the proposed Y-shaped architecture.
>
> **W3: Ablation on the number of shared vs. task-specific layers**
>
> We have added the ablation results in the table below and in Table 2 of the revised paper.
>
> | Number of shared layers ($M$) | MME ↑ | POPE ↑ | GenEval ↑ |
> | --- | --- | --- | --- |
> | 8 | 1368 | 82.9 | 0.28 |
> | 12 | 1402 | 84.5 | 0.29 |
> | 16 | 1322 | 84.0 | 0.25 |
>
> We experiment with different numbers of shared layers while keeping the training settings identical. Sharing 12 layers (half of the transformer depth) achieves the best performance across both understanding and generation tasks. This suggests that a moderate level of sharing facilitates cross-task representation learning. However, in deeper layers, the objectives of understanding and generation progressively diverge; excessive sharing in these later stages introduces task interference, ultimately degrading performance.
>
> **Minor presentation issues**
>
> Thank you for pointing these out. We have corrected all the issues in the revised version.
>
> **Q1: Is a symmetric branch necessary? Would asymmetric branches (e.g., larger generation branch) work as well or better?**
>
> Symmetric branch is not necessary, and we used asymmetric branches in our main experiment.
>
> In the ablation setting, the vision head contains 0.07B parameters, and the generation-specific transformer layers have 0.43B parameters (Table 1). In the main experiments, we adopt the asymmetric branches — we enlarge the vision head and expand the generation parameters to 0.76B parameters (Table 4 and Table 5), together with more generation data. This leads to 39% and 35% improvements on the GenEval and MJHQ benchmarks respectively.

---

### Author Response · Authors · 2025-11-27
**General Response**

**Dear ACs and Reviewers,**

Before entering the summary, we would like to note that Reviewer 5cnP had already **updated the score** from 2→4 during the discussion phase based on partial rebuttal feedback. As the discussion was closed early, further clarifications could not be exchanged.

Thank you for your insightful comments and suggestions. Building a unified model that performs well on both multimodal understanding and generation remains highly challenging, particularly when compared with systems trained on large proprietary datasets, substantial compute and technical optimization. Given this context, our goal is not to build a fully SOTA unified model—which would require considerable additional high-quality training data and compute—but to propose and validate a more effective architecture under realistic research resources.

### **Contributions**

The main contributions recognized by reviewers are as follows:

- **Insightful analysis**

    Reviewers highlighted that our systematic study of modality alignment dynamics across understanding and generation tasks provides “new” and “insightful” findings (a4kZ, 5cnP, P6ZU). In particular, identifying the divergent alignment behaviors—monotonically increasing for understanding versus rise-and-fall for generation—fills an important gap in prior multimodal research (P6ZU).

- **Novel and well-motivated architecture**

    Reviewers noted that our Y-shape design “offers a novel way” to resolve the conflict between alignment patterns (a4kZ, k4Sk). The shared early layers and task-specialized branches were recognized as a clear and well-motivated architectural choice that directly operationalizes our alignment insights (k4Sk, P6ZU).

- **Strong empirical performance and validated design choices**

    Reviewers emphasized that UniFork achieves consistently superior results across benchmarks compared with ablations and prior unified models (a4kZ, k4Sk). They also appreciated the breadth of experiments and the careful documentation of training configurations that strengthen the empirical evidence (5cnP).


### **New Clarifications and New Experiments**

In addition to the pointwise responses below, we summarize supporting clarifications and experiments added in the rebuttal according to the reviewers’ suggestions.

**New Experiments and Analysis**

- Ablation study on the depth of the shared vs. task-specific layers (*M* and *N*) in Table 2. [a4kZ, k4Sk, 5cnP, P6ZU]
- Scalability validation on model size in Table 6. [k4Sk, 5cnP, P6ZU]

**New Clarifications about comprehensive baselines and benchmarks**

- While we acknowledge the value of expanding benchmark and baseline coverage, such scaling requires much more high-quality data and compute beyond the scope of this work. We believe the current evaluation suite supports and validates the proposed architecture effectively.
- Moreover, we note that a **subsequen work** [1] adopts and further scales the Y-shaped architecture—placing task-specific layers only in deep stages while maintaining shared early representations. With substantially larger training compute, this work reports superior performance over a broader range of baselines (BAGEL, Janus-Pro, show-o2, Emu3, TokenFlow), consistent with those suggested by Reviewers a4kZ, k4Sk, 5cnP, and P6ZU. It is also evaluated on wider understanding and generation benchmarks. Their results provide external evidence that UniFork is capable of reaching SOTA performance under more extensive training regimes.

We hope our pointwise responses below can clarify remaining confusion and address the concerns raised. Thanks to all reviewers, and we have incorporated some mentioned contents into the revised paper, highlighted in **blue**.

### **Reference**

[1] Shen, Tao, et al. "MammothModa2: A Unified AR-Diffusion Framework for Multimodal Understanding and Generation." *arXiv preprint arXiv:2511.18262* (2025).

---

### Meta-Review · Area_Chair_SWcW · 2025-12-26

**Summary:**

Several reviewers raised overlapping major concerns, summarized as follows:

1. Limited benchmarking on both the understanding tasks and text-to-image (T2I) generation benchmarks beyond GenEval (raised by two reviewers).

2. Outdated or incomplete baseline comparisons, with missing comparisons to more recent or stronger models (raised by all reviewers).

3. Insufficient experimental scale, which limits the strength and generalizability of the empirical conclusions.

4. Lack of verification or ablation studies to quantify the impact of the different number of shared layers (shared by all reviewers).

**Reviewer Concerns:**

The AC believes that the third and fourth concerns are reasonably addressed in the rebuttal, where the authors provide ablation results on the number of shared layers and report additional performance results based on a 7B model.

However, the first, and in particular the second, concerns remain insufficiently addressed. As acknowledged by the authors, expanding benchmark coverage and incorporating stronger or more recent baselines would require time beyond the rebuttal period as well as substantial additional resources.

**Reviewer Scores:**

`Reviewer a4kZ (initial score: 4)` is likely to increase their rating, as some concerns are partially addressed; however, a change to an acceptance recommendation is unlikely due to the remaining issues regarding benchmark coverage and baseline comparisons.

`Reviewer k4Sk (initial score: 4)` is similarly expected to raise their rating, but not to the level of acceptance for the same reasons.

`Reviewer 5cnP (initial score: 2)` may increase their rating toward a borderline reject or borderline accept, depending on how they weigh the rebuttal clarifications.

`Reviewer P6ZU (initial score: 6)` is likely to maintain their positive assessment.


Overall, the extrapolated ratings place the paper in a borderline range. While the observations and the proposed Y-shaped architecture are interesting and show potential, the unresolved concerns, particularly regarding insufficient benchmarking and lack of comparison with more recent baselines, indicate that the manuscript is not yet ready for publication. Addressing these issues would require additional time and more comprehensive empirical evidence that cannot be addressed within the rebuttal time frame.

---

### Decision · Program_Chairs · 2026-01-26

Reject